# Treating common mental disorder including psychotic experiences in the primary care improving access to psychological therapies programme (the TYPPEX study): protocol for a stepped wedge cluster randomised controlled trial with nested economic and process evaluation of a training package for therapists

Polly-Anna Ashford,[1] Clare Knight,[2] Margaret Heslin,[3] Allan B Clark,[4] Mona Kanaan,[5] Ushma Patel,[6] Freya Stuart,[2] Thomas Kabir,[7] Nick Grey,[8,9] Hannah Murray,[10] J Hodgekins,[4] Nesta Reeve,[11] Nicola Marshall,[6] Michelle Painter,[11] James Clarke,[6] Debra Russo,[2] Jan Stochl,[2,12] Maria Leathersich,[4] Martin Pond,[4] David Fowler,[13] Paul French,[14] Ann Marie Swart,[4] Mary Dixon-Woods ,[15] Sarah Byford ,[16] Peter B Jones ,[2] Jesus Perez [17,18]

For numbered affiliations see end of article.

**Correspondence to**
Professor Jesus Perez;
jp440@cam.ac.uk

## ABSTRACT

**Introduction** At least one in four people treated by the primary care improving access to psychological therapies (IAPT) programme in England experiences distressing psychotic experiences (PE) in addition to common mental disorder (CMD). These individuals are less likely to achieve recovery. IAPT services do not routinely screen for nor offer specific treatments for CMD including PE. The Tailoring evidence-based psychological therapY for People with common mental disorder including Psychotic EXperiences study will evaluate the clinical and cost-effectiveness of an enhanced training for cognitive behavioural therapists that aims to address this clinical gap.

**Methods and analysis** This is a multisite, stepped-wedge cluster randomised controlled trial. The setting will be IAPT services within three mental health trusts. The participants will be (1) 56–80 qualified IAPT cognitive behavioural therapists and (2) 600 service users who are triaged as appropriate for cognitive behavioural therapy in an IAPT service *and* have PE according to the Community Assessment of Psychic Experiences—Positive 15-items Scale. IAPT therapists will be grouped into eight study clusters subsequently randomised to the control-intervention sequence. We will obtain pseudonymous clinical outcome data from IAPT clinical records for eligible service users. We will invite service users to complete health economic measures at baseline, 3, 6, 9 and 12-month follow-up. The primary outcome will be the proportion of patients with common mental disorder psychotic experiences who have recovered by the end of treatment as measured by the official IAPT measure for recovery.

**Ethics and dissemination** The study received the following approvals: South Central—Berkshire Research

## STRENGTHS AND LIMITATIONS OF THIS STUDY

⇒ The Tailoring evidence-based psychological therapY for People with common mental disorder including Psychotic EXperiences study will provide the first multisite, stepped-wedge cluster randomised controlled trial to investigate the clinical and cost-effectiveness of an intervention in the UK improving access to psychological therapies (IAPT) programme.

⇒ The study intervention seeks to address the currently underserved needs of an important group of people accessing IAPT services.

⇒ The stepped-wedge cluster design allows the intervention to be rolled out over time to all potential participants, while reducing operational and logistical challenges.

⇒ The primary limitation is potential loss to follow-up, resulting in missing data for the secondary health economics outcome. This would challenge the internal validity of the conclusions drawn from the study.

Ethics Committee on 28 April 2020 (REC reference 20/SC/0135) and Health Research Authority (HRA) on 23 June 2020. An amendment was approved by the Ethics Committee on 01 October 2020 and HRA on 27 October 2020. Results will be made available to patients and the public, the funders, stakeholders in the IAPT services and other researchers.
**Trial registration number** ISRCTN93895792.

## INTRODUCTION

Psychotic experiences (PE), such as attenuated and/or fragmentary paranoid beliefs and hallucinations, are relatively common in the general population, especially among young people.[1] Though systematic reviews and evidence synthesis indicate that 30%–40% of those with intense and frequent attenuated psychotic symptoms will transition to a psychotic disorder,[2 3] studies including individuals with shorter and less intense mental symptoms have shown that far fewer (~10%) make such a conversion.[4 5] Nonetheless, PE still predict propensity to seek treatment from mental health services[6] and are a marker for severity of other, non-psychotic common mental disorders (CMD), particularly depression and anxiety.[4 7 8]

In England, the improving access to psychological therapies (IAPT) programme is the main provision for people with CMD. Offering UK National Institute for Health and Care Excellence (NICE)-approved psychological therapies, the programme is predominantly based on cognitive behavioural therapy (CBT). National standards specify that 50% of IAPT service users should achieve recovery, a key performance indicator defined by reduction in scores on indices of depression and anxiety. For the first time since its inception in 2008, IAPT met this target in 2016–2017, with 51% of people nationally being considered recovered by the end of treatment.[9]

Evidence suggests that at least one in four people treated by IAPT may have a common mental disorder that includes psychotic experiences (CMD-PE). These individuals are at increased risk of not demonstrating recovery.[5 10] Currently, IAPT neither screens for, nor offers specific treatments for CMD-PE. Treatment protocols focus exclusively on mood disturbance, leaving PE undetected and untreated. Given the complexity and comorbidity of CMD-PE, standard treatment interventions are likely to be suboptimal.

Though some effective psychological treatments for PE exist, such as CBT for at-risk mental states,[11 12] they are dispersed throughout various service settings, and as such are inadequate to assess or treat the whole condition of CMD-PE. An innovative National Institute for Health Research (NIHR)-funded programme—Tailoring evidence-based psychological therapY for People with common mental disorder including Psychotic EXperiences (TYPPEX)—is targeting this clinical gap. It seeks to adapt and assemble available treatment options for CMD-PE into a practical therapeutic toolbox for cognitive behavioural therapists in IAPT known as CBT tailoring for severity (CBT-ts), rolled out via an enhanced training package and supervision programme.

A single-arm, three-site feasibility study (National Health Service (NHS) Research Ethics Committee (REC) reference 19/SC/0077) demonstrated the successful delivery of CBT-ts to 31 therapists and outcome data collection for 153 Community Assessment of Psychic Experiences (CAPE) +patients on their caseload, and provided qualitative information used to refine the training intervention (study report available by request from the authors).

This protocol paper describes the TYPPEX stepped wedge cluster randomised controlled trial (TYPPEX swcRCT), which aims to determine the clinical and cost-effectiveness of CBT-ts versus IAPT standard care (SC) in IAPT service users with CMD and also PE, as determined by the CAPE—Positive 15-items Scale (CAPE-P15).[13] This pragmatic trial will be the first to identify and treat people with CMD-PE in IAPT services. It will provide valuable evidence for the development of interventions for this important and underserved population, while adding to the body of knowledge about design and conduct of randomised controlled trials in the IAPT setting.

## METHODS

This protocol is reported in accordance with Standard Protocol Items: Recommendations for Interventional Trials guidance[14] and refers to study protocol V.2.0 dated August 2020, which was the first version to be implemented. Amendments made to the protocol since commencement of the trial are shown in table 1. Participant timelines and measures are summarised in table 2.

| Table 1 | Summary of ethical amendments |
|---|---|
| **Protocol** | **Primary reasons for amendment** |
| 2.0, 04/08/2020 | ► Addition of secondary clinical outcomes (Reliable Recovery and Reliable Improvement).<br>► Addition of retrospective pseudonymous clinical data collection from the beginning of UK lockdown due to COVID-19, until randomisation.<br>► Intervention adaptation: training to be delivered either online or face-to-face.<br>► Changes to the service user consent process: moving from full consent on tablet devices to consent to contact followed by full consent online at home. |
| 3.0, 12/05/2021 | ► Video conferencing, eg, Zoom included as an option for process evaluation interviews.<br>► Removal of restriction of only teams from the same NHS Trust merging to form a cluster. |

**Table 2** Schedule of IAPT user enrolment, interventions and assessments

| Time point | Study period (control and randomised conditions) | | | | | | |
| --- | --- | --- | --- | --- | --- | --- | --- |
| | Screening | Baseline | Treatment | Follow-up | | | |
| | -t1** | -t2 | Up to 20 sessions | 3 | 6 | 9 | 12 |
| **Enrolment** | | | | | | | |
| Eligibility screen | x | | | | | | |
| Consent to contact (for health economics measures only) | | | | | | | |
| Informed consent (for health economics measures only) | | x | | | | | |
| IAPT CBT standard care | | | ▬▬▬▬▬ | | | | |
| **Assessments** | | | | | | | |
| PHQ-9† | x | x | X | | | | x |
| GAD-7/ADSM† | x | x | X | | | | x† |
| EQ-5D-3L | | x | | x | x | x | x |
| EQ-5D-5L | | x | | x | x | x | x |
| EI-ADSUS | | x | | x | x | x | x |
| CAPE P-15 | x | | | | | | x |

*The duration between the screening visit and baseline is anticipated to be 1–3 weeks, depending on typical frequency of therapy sessions in IAPT service. During this time the CAPE-P15 is scored and eligibility is confirmed. CAPE+IAPT users are approached to participate in the health economics data collection during the baseline visit. Clinical data are obtained for all CAPE+IAPT users unless they opt out.

†PHQ-9 and GAD-7 are routine IAPT clinical measures and are collected during every clinical contact. At 12-month follow-up, these measures will be collected via an opt-in process to provide additional data.

ADSM, anxiety disorder specific measure ; ADSUS, adult service use schedule; CAPE-P15, Community Assessment of Psychic Experiences—Positive 15-items Scale; CBT, cognitive behavioural therapy ; GAD-7, General Anxiety Disorder Assessment-7; IAPT, improving access to psychological therapies ; PHQ-9, Patient Health Questionnaire-9.

## Public and patient involvement

An experienced public and patient involvement (PPI) representative contributed significantly to the development of the PPI strategy set out in the grant application and continues to have study oversight as a member of the steering committee. The submission was endorsed by a co-applicant with relevant lived and academic experience and a PPI coordinator with lived experience was employed to facilitate PPI activity.

At the planning stage a Lived Experience Advisory Panel ('LEAP') was convened comprising 11 people with lived experience including diverse community members. The LEAP influenced some research questions extending the scope of a systematic review within the research programme[15] and informing qualitative interview questions.

Since TYPPEX is a pragmatic trial, the study design is largely dictated according to the way that IAPT services operate. The primary outcome measure is a national metric of 'recovery' derived from routinely collected IAPT data, and is not open to alteration. LEAP members positively influenced secondary outcome measures including introducing the CAPE-P15 at final follow-up.

LEAP involvement in the development of the intervention (CBT-ts) was highly influential, for example, in review of the training resources and ideas about language and delivery. All patient-facing resources are produced in partnership with the LEAP, who have been invaluable in raising issues around acceptability and accessibility. The panel reviewed the feasibility study protocol and gave feedback on adaptations to remote working for the main trial. All ethical submissions undergo thorough PPI review with strong focus on acceptability of consent processes.

PPI representation from IAPT services was not initially possible since primary care services do not routinely screen for PE. However, after the trial's feasibility study, we have expanded involvement to include three people recruited for such study, who have lived experience that matches our target population. We have also expanded involvement to include four IAPT CBT therapists, also drawn from the feasibility study, to ensure acceptability and usability of CBT-ts and research methods within an IAPT setting, together contributing an additional seven perspectives.

People with lived experience will be involved in qualitative data interpretation to provide context and sense checking of themes and anonymised service-user quotes as part of the process evaluation. LEAP members have coauthored papers and presented posters at public events and dissemination activity will continue to be informed and enhanced by people with lived experience.

PPI will remain reflective and responsive to new insights and the needs of the study, working to the NIHR INVOLVE's standards for public involvement.[16]

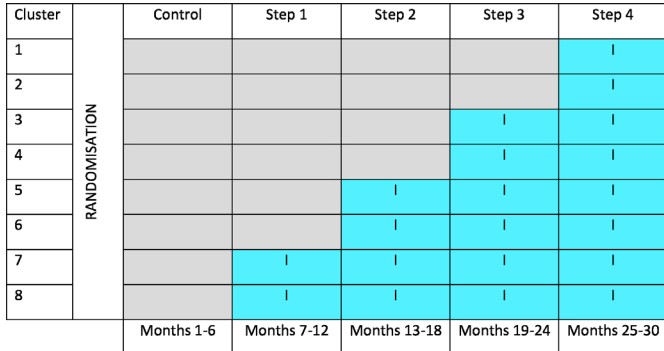

| Cluster | | Control | Step 1 | Step 2 | Step 3 | Step 4 |
|---|---|---|---|---|---|---|
| 1 | | | | | | I |
| 2 | | | | | | I |
| 3 | R | | | | I | I |
| 4 | A | | | | I | I |
| 5 | N | | | I | I | I |
| 6 | D | | | I | I | I |
| 7 | O | | I | I | I | I |
| 8 | | | I | I | I | I |
| | | Months 1-6 | Months 7-12 | Months 13-18 | Months 19-24 | Months 25-30 |

Key:

| Control phase |
|---|
| Intervention phase (I) |

**Figure 1** Stepped wedge trial diagram.

## Study setting

The IAPT programme in England began in 2008 with the aim of improving access to evidence-based psychological treatment for CMD, particularly depression and anxiety. It sought to increase public access to NICE-approved psychological therapies, predominantly CBT, by offering flexible referral routes, including self-referral. The programme has continued to expand over time and currently assesses over 1.3 million people with CMD annually, delivering therapy to approximately 550 000.[9] NHS England has committed to further expansion of the programme, aiming to see 1.9 million patients annually by 2024.[17]

The programme provides treatment for CMD for people aged 17 or over. Its pathways involve a stepped care model, providing steps 2 and 3 of the NICE five-step approach to the treatment of CMD. At step 2 in the IAPT programme, large numbers of service users, with less severe conditions may be allocated to a single therapist. These service users may receive a variety of interventions, including brief face-to-face therapy, telephone support, computerised CBT or guided self-help. At step 3, highly trained CBT therapists manage smaller caseloads of people with more severe CMD. These therapists use more intensive treatment methods, delivering up to 20 individual face-to-face psychological therapy sessions. This clinical setting is immediately below psychiatric treatment by secondary mental health services (NICE step 4).

Step 3 IAPT CBT therapists will have completed a 1 year postgraduate diploma (PG Dip) in CBT accredited by the British Association for Behavioural and Cognitive Psychotherapies (BABCP). Entry requirements for the PG Dip include a secured work placement for clinical practice and a BABCP recognised core profession, such as mental health nursing or counselling psychology.[18]

## Trial design and participants

The TYPPEX study is a pragmatic, multicentre swcRCT with nested health economic and process evaluations. It involves eight NICE step-3 IAPT teams (clusters) of CBT therapists across three mental health trusts in England: Cambridgeshire and Peterborough NHS Foundation Trust, Norfolk and Suffolk NHS Foundation Trust and Sussex Partnership NHS Foundation Trust. Each trust (organisational unit within the National Health Services of England and Wales) serves a mixed urban and rural population with a wide socioeconomic range, including some high levels of deprivation, covering a total population of approximately 2.8 million.

Clusters will be randomised to different sequences rather than, strictly speaking, different arms. The sequences dictate the order in which each cluster switches from the control to intervention condition. TYPPEX swcRCT has five measurement periods and four steps, with two clusters switching to the intervention condition at each step. As the intervention is cluster randomised at the level of IAPT team, and the delivery of therapy by these teams may be subject to regional variation, each cluster will act as its own internal control. All clusters will start in the control condition and move to the intervention condition (see figure 1).

The primary outcome of this study will be the proportion of eligible individuals with CMD-PE, as determined by the CAPE-P15 cut-off of 1.47,[19] who have recovered by the end of treatment, measured via pseudonymous clinical outcome data obtained from IAPT clinical records and IAPT-defined metrics of recovery, as specified in table 2table 3

## Eligibility criteria

IAPT therapist inclusion criteria are: (1) qualified IAPT CBT therapist, and (2) willing and able to provide informed consent to receive CBT-ts training and supervision. IAPT therapist exclusion criteria are: (1) does not deliver CBT therapy in current role, (2) works across more than one locality IAPT team (to avoid potential contamination eg, via team meetings, shared office space) and (3) participated in an earlier TYPPEX feasibility study.

Service user participants are those individuals accepted onto the NICE step-3 IAPT caseload for participating therapists and meeting the study eligibility criteria, which are: (1) accepted onto the IAPT caseload for therapy and therefore meet service-specific inclusion criteria to access IAPT treatment (irrespective of whether service has been accessed previously or not), (2) meet current criteria for IAPT CBT treatment and (3) assessed for PE, according to the CAPE-P15 questionnaire in their clinical record. Additional inclusion criteria for health economics substudy only: (4) presence of PE according to the CAPE-P15 score, (5) in the judgement of the treating therapist has sufficient proficiency in English to complete research questionnaires and (6) able and willing to provide written informed consent.

IAPT service user exclusion criteria are: (1) presence of mental disorder, such as complex and severe depression, based on standard IAPT assessment meriting routine referral to NICE step-4 treatment, that is, to secondary mental health services.

## Participant withdrawal

The right of any participant—therapist or service user—to discontinue participation without giving any reason will be respected. People will remain free to withdraw from the trial at any point without prejudicing their further treatment or, for therapists, employment. Reason for withdrawal, if given, will be recorded. Service users who have refused or withdrawn consent for data sharing will be excluded from pseudonymous routine data collection.

## Randomisation

IAPT therapists from local teams within participating sites will be recruited and grouped as study clusters before randomisation (without stratification by cluster characteristics) to the control-intervention sequences via computer written code. In most cases, clusters will be formed from a single IAPT team. Some teams may have fewer IAPT CBT therapists available for training due to empty therapist posts or service demands. In these cases, two local IAPT teams will be merged to create a cluster. Once clusters are allocated, IAPT service managers will be approached and CBT-ts training courses arranged in line with each cluster randomisation point.

## Blinding

The study will be conducted in a controlled open format with unblinded assessment; neither the research team nor the therapists that form the study clusters can be blind to the intervention. IAPT service users will be made aware of the study via an information leaflet given during screening but will not know whether their therapist has received CBT-ts training.

## Sample size

CBT-ts training will be delivered to step-3 CBT therapists in eight clusters, randomised at four steps. In each cluster, approximately 7–10 therapists will receive training, with the intention for at least five therapists to deliver therapy throughout each cluster period, to allow for inevitable therapist turnover and sickness.

Based on our prevalence studies of CMD-PE in IAPT[5 10] and considering average caseloads of 15–20 services users per therapist over 6 months, each therapist will deliver therapy to a minimum of three people. Pragmatically, the total number of eligible service users receiving therapy from a participating therapist during the whole trial may be higher, but only control service users who complete therapy during their cluster's control phase and intervention service users who complete therapy prior to the end of the intervention phase, will be included in the primary outcome analysis. Based on results from our previous TYPPEX feasibility work, we expect that 6-month steps would be sufficient for this.

All service users will be identified as having PE with the CAPE-P15 score threshold of 1.47[19] for both frequency of and distress associated to psychotic experiences at baseline, resulting in a minimum of 15 service users per cluster period. Eight clusters and four randomisation steps result in 20 control condition cluster periods and 20 intervention condition cluster periods. Three hundred IAPT service users will be recruited during the control condition and 300 during the intervention condition across all sites. In addition to the number of clusters and the steps presented above, we have assumed: an intracluster correlation coefficient of 0.05, a significance level of 0.05 and an improvement in recovery from 0.39 (control) to 0.58 (intervention).[5 10] Based on these assumptions, the proposed sample size will allow us to detect such a difference with just over 80% power. Although this ignores that clustering also happens at the therapist level, it has been shown that ignoring an additional, lower level of clustering is a conservative estimate of the power. This is not sensitive to the choice of intracluster correlation coefficient.[20]

Also during the feasibility work we found that approximately 50% of the sample providing pseudonymised routine data would consent to complete the health economic measures. Thus, we aim that 150 service users will be consented to complete health economic measures during the control condition and 150 during the intervention phase across all sites.

## Procedures

All IAPT service users who begin a course of CBT from the outset of the trial will be asked to complete a CAPE-P15 screening questionnaire and will receive a leaflet during their first treatment session. The leaflet explains that the trial is taking place in the service, and if eligible, their pseudonymous clinical data will be used for the research. Awareness of the trial is also raised in participating sites using posters. Both the poster and leaflet provide service users with information on the local policy for how they can opt out should they wish their data not to be used for research purposes. The CAPE-P15 screening questionnaire is currently accessible for clinical information as part of routine service use for participating sites.

IAPT service users with PE according to the CAPE-P15 will be asked by their therapist to provide consent to be contacted about participating in the health economic data collection, since this data cannot be collected from clinical records. If consent is obtained, the service user will be sent a participant information sheet and an electronic consent form by email (or paper versions by post) prior to the following treatment session, followed by baseline health economics measures.

Recruited service users will receive standard IAPT care from a participating therapist (control condition) or standard IAPT care from a participating therapist following CBT-ts training (intervention condition). They will be contacted by email or post for follow-up health economics data collection at 3, 6, 9 and 12 months after baseline.

## Intervention

### CBT-ts

CBT-ts is a team-level service improvement intervention. It aims to enhance existing CBT skills of NICE step-3

IAPT CBT therapists and is not a novel therapeutic intervention. Therefore, it is important to highlight that all IAPT service-users receiving CBT under the care of a CBT-ts trained therapist, including those who do not have CMD-PE, may be exposed to the effects of the training.

CBT-ts is based on the CBT framework currently used nationally in IAPT but includes adaptations to existing interventions to specifically address PE. The training takes a knowledge, skills and attitude focus enhancing existing knowledge of CBT. It includes three modules delivered online as workshops supplemented by a blended learning package with access to an online network and provision of written resources. Therapists are provided with a single pack of editable templates, which they may or may not use in their clinical practice.

The structured training programme includes:

Module 1:

► Introduction to the concept of CMD-PE and the relationship between CMD and PE.
► Prevalence and impact in IAPT services.
► Identification of PE.
► Normalisation and validation of CMD-PE.

Module 2:

► Incorporating assessing PE into existing assessments within CBT.
► Suitability for IAPT.
► Understanding CMD-PE maintenance cycles.
► Approaches to the therapeutic conceptualisation and formulation of CMD-PE including comorbidity.

Module 3:

► Adaptations to existing cognitive behavioural strategies and the application of these to promote the therapeutic change, using change methodologies, such as metaphors, exposure, cognitive restructuring and behavioural experiments.
► Relapse prevention.

Training is supported by six CBT-ts supervision sessions, which commence 1 month post training and are subsequently delivered monthly, either online or face-to-face. Following completion of training, CBT therapists will identify people with CMD-PE according to the CAPE-P15 and treat them in line with current CBT interventions, incorporating their enhanced skills, as necessary. Treatment duration is non-prescriptive and expected to range between 8 and 14 sessions dependent on therapists' clinical decisions. Treatment duration is not expected to exceed 20 weekly sessions, in line with current standard guidance in IAPT services, nationally.

## SC

The control comparator is standard CBT care delivered within IAPT services prior to CBT-ts training. CBT is delivered as part of range of interventions offered at step 3 within the NICE stepped care model. CBT treatment protocols are delivered in accordance with NICE guidance for people with CMD. Treatment duration varies according to presenting problem and minimum dose necessary to achieve recovery, but NICE recommends individuals should receive up to 20 weekly sessions unless they recover beforehand (https://www.nice.org.uk/guidance/cg123/chapter/1-guidance).

## Outcomes

A summary of study outcomes is shown in table 3, and schedules of outcome measures for service users and therapists are provided in tables 2 and 4.

## Process evaluation

The aim of the process evaluation is to assess the views of all stakeholders involved in the experience and delivery of the TYPPEX swcRCT and CBT-ts, and to investigate the implementation of CBT-ts as designed, including any influences on uptake, delivery and fidelity. An overview of the key themes explored with each stakeholder group is given in table 5.

The aim of sampling for the process evaluation will be to achieve theoretical saturation,[21] where further new data are unlikely to achieve further insights or to add to analytic depth. Up to 32 service users across the eight clusters with scores of 1.47 and above on the CAPE-P15 and who have completed at least two CBT treatment sessions with a CBT-ts trained therapist will be recruited including those who have completed treatment, those who are still undergoing therapy and those who dropped out of treatment. Where possible, purposive sampling will be used to ensure diversity of gender and ethnicity and exposure to number of sessions of therapy.

Between 16 and 24 IAPT therapists (with a minimum of two per cluster) who completed CBT-ts training and subsequently delivered at least three treatment sessions to a minimum of two service-users scoring 1.47 and above on the CAPE-P15 will be recruited. Additionally, we will conduct 18–25 interviews with IAPT clinical supervisors/managers from participating IAPT teams who line manage or supervise a CBT-ts trained therapist, senior non-clinical NHS managers with responsibility for delivery of IAPT services at Trust level or responsibility for commissioning IAPT services, and members of the trial team and collaborators who contributed to the design, development or implementation of TYPPEX. In all cases, purposive sampling will be used to ensure representation of a wide range of perspectives and experiences.

## Data management

Pseudonymised outcome data will be received by Norwich Clinical Trials Unit from sites and uploaded to a central study database stored on servers based at the University of East Anglia. Health economic questionnaire data will be collected using Research Electronic Data Capture (REDCap) software (https://www.project-redcap.org). Questionnaires will be completed online or by post by service users. Pseudonymised clinical data will be linked to consented participants' health economics data held by the Norwich Clinical Trials Unit (NCTU).

Data collection will include retrospectively recorded IAPT data from 23 March 2020 until trial start. This

**Table 3** Summary of study outcomes

| Category | Measure | Primary/secondary | Details |
|---|---|---|---|
| Service user recovery | IAPT-defined 'Recovery' | Primary | The primary outcome is the proportion of IAPT service users with CAPE-P15 score of 1.47 or above who have recovered by the end of treatment. Recovery is a national IAPT programme performance metric, with an individual deemed recovered if they scored above the clinical cut-off on the PHQ-9[35] and/or GAD-7[36] before treatment, ie, 10 and 8, respectively, and below the clinical cut-off at the end of treatment on both measures.[37] If recorded, an anxiety disorder specific measure is used in place of the GAD-7 if that is the focus of treatment.[38] |
| | IAPT-defined 'Recovery' | Secondary | As for the primary outcome, but for IAPT service-users reaching the lower threshold score for CMD-PE (CAPE-P15 score of 1.30 and above).[39] |
| | IAPT-defined 'Reliable Improvement' | Secondary | Service users are considered reliably improved if they show any improvement in scores on the appropriate outcome measures between pre and post treatment, that exceeds the measurement error of the scales.[38] |
| | IAPT-defined 'Reliable Recovery' | Secondary | Service users are considered reliably recovered if they meet both criteria for Reliable Improvement and for Recovery.[38] |
| Health economic measures | Modified adult service use schedule for early intervention (EI-ADSUS) | Secondary | Individual-level resource use will be measured using a modified version of EI-ADSUS. This was developed in previous research with similar populations[37] and adapted for electronic participant self-completion. It will include all-cause hospital and community-based health and social care resource use as well as information on time off work due to mental health. |
| | EuroQol measure of health-related quality of life three level (EQ-5D-3L) | Secondary | Health-related quality of life will be measured using the EQ-5D-3L, a generic, preference-based measure based on five dimensions (mobility, self-care, usual activities, pain/discomfort and anxiety/depression). Each dimension is rated on three levels (no problems, some problems and severe problems).[40] |
| | EuroQol measure of health-related quality of life five level (EQ-5D-5L) | Secondary | The EQ-5D-5L is also a health-related quality of life measure based on five dimensions (mobility, self-care, usual activities, pain/discomfort and anxiety/depression), but each dimension is rated on five levels (no problems, slight problems, moderate problems, severe problems and extreme problems.[41] It has been demonstrated that the EQ-5D-3L is a useful, valid instrument in young people with emerging psychotic symptoms.[42] However, the new five-level version, may prove more sensitive to change and is now recommended for economic evaluation by National Institute for Health and Care Excellence[43] but has not been validated in this population. Therefore, we will test the sensitivity of the EQ-5D-5L in comparison to the EQ-5D-3L. |
| Therapist adherence | Supervision checklist | Secondary | Therapist adherence measured using a supervision checklist and adherence score completed during the 6 monthly supervision sessions held after CBT training. Trial supervisors will review and collate therapy session notes and rate adherence to the CBT-ts approach. |

Continued

| Table 3 | Continued | | |
|---|---|---|---|
| **Category** | **Measure** | **Primary/secondary** | **Details** |
| CAPE-P15, Community Assessment of Psychic Experiences—Positive 15-items Scale ; CBT, cognitive behavioural therapy ; CBT-ts, CBT tailoring for severity ; CMD-PE, common mental disorder psychotic experiences ; GAD-7, General Anxiety Disorder Assessment-7; IAPT, improving access to psychological therapies ; PHQ-9, Patient Health Questionnaire-9. | | | |

data, covering the period of UK lockdown during the COVID-19 pandemic, will provide information on the nature of IAPT service provision during this time, and any changes in baseline morbidity. The aim of the data collection is to provide insight into standard of care and baseline severity during the pandemic, to assist decision-making by the trial oversight committees.

Data will be handled in accordance with a data management plan and comply with the principles of the International Conference on Harmonisation Good Clinical Practice, within the Standard Operating Procedures for Data Management in NCTU and where appropriate with the University of East Anglia Information Technology (UEA IT) procedures. Interview transcriptions will be anonymised in adherence with the 'Guidance on Anonymisation' issued by the UK Data Service (https://www.ukdataservice.ac.uk/manage-data/legal-ethical/anonymisation.aspx). Anonymised data will be stored in password-protected files on the firewall-protected University of Cambridge servers.

### Statistical analyses

The primary analysis will compare IAPT-defined recovery of service users with CMD-PE as determined by the CAPE-P15 cut-off of 1.47 in the control phase with those in the intervention phase (post-training). Recovery proportions and means will be compared at individual therapist and team level after adjusting for the stepped-wedge cluster design. Adjustment will also be made for the baseline characteristics and other covariates. Time will be included as a set of indicator variables.

Analyses of clinical effectiveness will be based on the intention-to-treat (ITT) population with all available follow-up data from all new service users with CMD-PE on the caseload of participating therapists, controlling for baseline scores on depression and anxiety disorder specific measures. The primary ITT analysis is intended to provide inferences regarding the effectiveness of the intervention overall. It does not provide inferences regarding the causal effect of the intervention itself, but on the intervention as deployed in 'real life', therefore compliance information is not necessary to ensure that the ITT analysis is valid. As soon as a therapist has their first session of CBT-ts training, they begin to practise differently, so this date will mark the beginning of their intervention phase.

A logistic mixed effects regression model will be used for the primary outcome analysis. The cluster effect by both the therapist team and therapist will be considered in the multilevel analysis. Adjustment for potential prognostic variables will be agreed prior to analysis between the trial statisticians and the chief investigator.

Intracluster correlation coefficients and 95% CIs will be reported for the primary and secondary outcomes. All analyses will be two-tailed and at the 5% level of significance.

Summary statistics of baseline characteristics at the cluster and the individual level will be reported where appropriate. Analyses will be conducted in Stata or R. Publication of results will include a participants/cluster flow diagram, and results will be reported according to

| Table 4 | Schedule of IAPT therapist enrolment, interventions and assessments | | | |
|---|---|---|---|---|
| | **Trial set-up** | **Randomisation** | **Control condition** | **Intervention condition** |
| **Identification** | | | | |
| IAPT teams and potential therapists identified and grouped to form clusters | x | | | |
| **Enrolment and randomisation** | | | | |
| Informed consent | x | | | |
| Clusters randomised | | x | | |
| **Intervention** | | | | |
| CBT-ts training | | | | x |
| CBT-ts supervision | | | | x |
| **Assessments** | | | | |
| Adherence and engagement record | | | | x |
| CBT-ts, cognitive behavioural therapy-tailoring for severity ; IAPT, improving access to psychological therapies . | | | | |

**Table 5** Key themes explored in the process evaluation for each stakeholder group

| Stakeholder group | Key themes |
|---|---|
| Service users | ▶ Experiences of treatment and therapist.<br>▶ Views on improving IAPT services.<br>▶ Views on CBT-ts. |
| Therapist | ▶ Design and goals of CBT-ts.<br>▶ Experience of the CBT-ts training.<br>▶ Experience of delivery and fidelity. |
| Line managers | ▶ Design and goals of CBT-ts.<br>▶ Experience of the CBT-ts training.<br>▶ Experience of delivery and fidelity at the therapist-service user level.<br>▶ Experience of delivery and fidelity at the IAPT service level. |
| Study team and collaborators | ▶ Rationale for CBT-ts.<br>▶ Implementation and components.<br>▶ Outcomes and measurement.<br>▶ Modifications and changes to CBT-ts. |
| Senior National Health Service managers/commissioners | ▶ Relevance of CBT-ts in IAPT.<br>▶ Implementation of CBT-ts model.<br>▶ Negotiating change.<br>▶ Measuring success. |

CBT-ts, cognitive behavioural therapy-tailoring for severity ; IAPT, improving access to psychological therapies .

the Consolidated Standards of Reporting Trials statement for stepped wedge designs.

### Economic analysis

The economic evaluation will be conducted covering the period from baseline to 12-month follow-up and will take the NHS/personal social services perspective preferred by NICE.[22] The intervention will be costed taking a bottom-up (micro-costing) approach[23] using data collected on face-to-face contacts from IAPT records and accounting for the ratio of direct face-to-face to indirect non-face-to-face time. Data on indirect time, including preparation and supervision, will be collected directly from the therapists. Details of resources required to deliver the CBT-ts training and supervision will be provided by the TYPPEX research team. Use of all other health and social care services will be collected via a modified version of the adult service use schedule for early intervention (table 1) collected using REDCap data collection software at baseline (covering the last 3 months) and 3, 6, 9 and 12 months (covering the period since the last interview). Resource use data will be combined with nationally applicable unit costs.[24 25] Productivity losses (applicable to sensitivity analyses only) will be calculated using the human capital approach by multiplying days off work for mental health reasons by the national average wage rate.[26]

Our primary economic analysis will be a cost-utility analysis using quality adjusted life years (QALYs) derived from the EuroQol measure of health-related quality of life (EQ-5D) based on complete case data. The EQ-5D-5L has been proposed as being more sensitive to change in mental health populations compared with the EQ-5D-3L.[27] However, there is little evidence to support this hypothesis. Therefore, we will compare the psychometric properties of the 3L and 5L versions. The version of the EQ-5D used in this analysis will be dependent on analyses comparing the two. QALYs will be calculated by applying appropriate utility weights to EQ-5D health states[28] and using the total area under the curve approach with linear interpolation between assessment points.[29] A secondary analysis will additionally explore cost-effectiveness using the primary outcome measure: IAPT-defined recovery.

Costs and outcomes will be compared at the final follow-up point and presented as mean values by trial phase with SD. Mean differences in costs and 95% CIs will be obtained by non-parametric bootstrap regressions to account for the non-normal distribution commonly found in economic data.[30] Incremental cost-effectiveness ratios (ICERs) will be calculated. Uncertainty will be explored using cost-effectiveness planes and cost-effectiveness acceptability curves based on the net-benefit approach.[31] These curves are an alternative to CIs around ICERs and show the probability that one intervention is cost-effective compared with the other for a range of values that a decision maker would be willing to pay for an additional unit of an outcome. To provide more relevant treatment-effect estimates, all economic analyses will include adjustment for stepped-wedge cluster design, relevant baseline characteristics and other covariates,[32] in line with the clinical analyses.

Sensitivity analyses will be conducted to explore the impact of missing data (using multiple imputation) and to explore cost-effectiveness from a broader perspective, including productivity losses because of time off work due to illness. Additionally, to assess the impact of missing data, we will compare those with economic data to the full sample included in the main clinical analysis to examine

any potential biases in terms of demographic and clinical factors.

## Process evaluation analysis

Analysis of process evaluation data will be based on the constant comparative method.[33][34] Practically, this will be achieved through multiple levels of analysis, systematically comparing data on different questions and from different sources (quantitative and qualitative) to achieve thematically ordered synthesis. Data analysis will take place after each stage of data collection so the analysis from each stage can inform data collection in the next. Some data analysis will also take place in parallel. The data will initially be subject to open coding using a combination of pre-selected questions and sensitising constructs identified from the literature. Codes will be increasingly grouped into higher-order explanatory categories through comparison and refinement in rounds of discussion and sense making. A set of overarching thematic categories will be agreed and used as a framework for further, more deductive, coding across the whole data set.

## Trial status

The trial was opened to IAPT service user recruitment in March 2021, with the first randomisation point scheduled for September 2021. Recruitment will continue until September 2023. The final 12-month follow-up is expected to be completed by October 2024.

## ETHICS AND DISSEMINATION

This study was approved by the South Central—Berkshire REC (REC reference 20/SC/0135 on 28 April 2020 and HRA on 23 June 2020). Subsequent amendments are shown in table 1.

The study is jointly sponsored by the University of Cambridge and Cambridgeshire and Peterborough NHS Foundation Trust. It was adopted onto the NIHR trial portfolio on 16 April 2020.

CBT-ts, including therapist training and supervision, is low risk for all participants. IAPT service users scoring above the CAPE-P15 thresholds for PE on whom data are being pseudonymously collected are under the care of the IAPT service for the duration of their participation and are subject to normal IAPT safety procedures. No specific risks, untoward incidents or adverse events related to CBT-ts are anticipated as the intervention aims to up-skill therapists rather than offering a new therapeutic intervention. This approach was supported by the results of a feasibility study as part of the wider TYPPEX research programme, during which no related serious adverse events were reported.

Safety outcome variables will be collected retrospectively each month from the routinely collected pseudonymous data. Aggregated safety information will be reported to the TYPPEX Data Monitoring and Ethics Committee throughout the randomisation condition. In addition, Principal Investigators, IAPT managers and

therapists will be requested to report reports relating to TYPPEX by eligible participants to the trial team.

The clinical data generated during therapy (CAPE-P15, General Anxiety Disorder Assessment-7 (GAD-7), Patient Health Questionnaire-9 (PHQ-9) and anxiety disorder specific measure) is recorded as part of routine care and is used by this trial via an opt-out consent process. This is designed to remove the burden on IAPT therapists with respect to recruitment, and to ensure a representative sample.

A subgroup of service users are consented via an opt-in process to provide additional data outside of the clinical setting. This includes cost and quality of life health economic data, and follow-up clinical data at 12 months (see table 2). This data will be linked with their IAPT clinical record.

Results from this trial will be presented at national and international conferences, including the Schizophrenia International Research Society Conference and IEPA *Early Intervention in Mental Health Conference*. Dissemination will also occur through the submission of a primary article on the outcomes and several subsequent articles considering the effects of the intervention stratified by service user and therapist variables. Primary outcomes will be disseminated more widely as part of a theory-driven implementation plan, developing a community of practise including patients, the public, economists and systems engineers experienced in care pathway design and evaluation, in addition to stakeholders in the IAPT services.

The training materials will be shared electronically and will be widely available to the NHS.

### Author affiliations
[1]Department of Clinical Psychology and Psychological Therapies, Norwich Medical School, University of East Anglia, Norwich, UK
[2]Psychiatry, University of Cambridge, Cambridge, UK
[3]King's Health Economics, King's College London Institute of Psychiatry Psychology and Neuroscience, London, UK
[4]Norwich Medical School, University of East Anglia, Norwich, UK
[5]Health Sciences, University of York, York, UK
[6]Cambridgeshire and Peterborough NHS Foundation Trust, Fulbourn, UK
[7]The McPin Foundation, London, UK
[8]Sussex Partnership NHS Foundation Trust, Worthing, UK
[9]Psychology, University of Sussex, Brighton, UK
[10]Psychology, University of Oxford, Oxford, UK
[11]Norfolk and Suffolk NHS Foundation Trust, Norwich, UK
[12]Kinanthropology, Charles University, Prague, Czechia
[13]Psychology, University of Sussex, Brighton, UK
[14]Manchester Metropolitan University, Manchester, UK
[15]THIS Institute (The Healthcare Improvement Studies Institute), University of Cambridge Primary Care Unit, Cambridge, UK
[16]Centre for the Economics of Mental and Physical Health, King's College London, London, UK
[17]Cambridgeshire and Peterborough NHS Foundation Trust, Cambridge, UK
[18]Medicine, Universidad de Salamanca, IBSAL, Salamanca, Castilla y León, Spain

**Contributors** PBJ and JP are co-CIs for and designed this trial. They also are joint senior authors for this manuscript. P-AA is the trial manager. CK was the manager for the wider research programme. UP led the design and development of the intervention, supported by DR. JH, NR, NM, JC, MP, NG, HM, DF and PF advised on the development of the intervention. TK leads on the PPI strategy. FS supports all

elements of the PPI strategy and, with DR, also delivers research assistant functions for the trial. ABC and MK contributed to the trial design and oversee the statistical analysis. JS advises on statistical matters. MH and SB lead the health economic component of the trial. MD-W oversees qualitative analyses and process evaluation. ML and MP are responsible for the trial data management. AMS oversees all operations at the Clinical Trials Unit. P-AA, CK, MH, ABC, MK and JP wrote the first draft of this manuscript. All authors reviewed, contributed to and commented on the final draft.

**Funding** This study is funded by the National Institute for Health and Care Research (NIHR) Programme Grant for Applied Research (PGfAR) RP-PG-0616-20003. JS, JP and PBJ are supported by the National Institute for Health and Care Research (NIHR) Applied Research Collaboration East of England (NIHR ARC EoE) at Cambridge and Peterborough NHS Foundation Trust. The views expressed are those of the authors and not necessarily those of the NIHR or the Department of Health and Social Care.

**Competing interests** None declared.

**Patient and public involvement** Patients and/or the public were involved in the design, or conduct, or reporting, or dissemination plans of this research. Refer to the Methods section for further details.

**Patient consent for publication** Not applicable.

**Provenance and peer review** Not commissioned; externally peer reviewed.

**ORCID iDs**
Mary Dixon-Woods http://orcid.org/0000-0002-5915-0041
Sarah Byford http://orcid.org/0000-0001-7084-1495
Peter B Jones http://orcid.org/0000-0002-0387-880X
Jesus Perez http://orcid.org/0000-0003-0740-190X

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
