## [Reviewer comments · BMJ Open]

ARTICLE DETAILS

TITLE (PROVISIONAL)	Treating common mental disorder including psychotic experiences in the primary care Improving Access to Psychological Therapies programme (the TYPPEX study): protocol for a stepped wedge cluster randomised controlled trial with nested economic and process evaluation of a training package for therapists
AUTHORS	Ashford, Polly-Anna; Knight, Clare; Heslin, Margaret; Clark, Allan; Kanaan, Mona; Patel, Ushma; Stuart, Freya; Kabir, Thomas; Grey, Nick; Murray, Hannah; Hodgekins, J; Reeve, Nesta; Marshall, Nicola; Painter, Michelle; Clarke, James; Russo, Debra; Stochl, Jan; Leathersich, Maria; Pond, Martin; Fowler, David; French, Paul; Swart, Ann Marie; Dixon-Woods, Mary; Byford, Sarah; Jones, Peter; Perez, Jesus

VERSION 1 – REVIEW

REVIEWER	Sequeira, Carlos CINTESIS
REVIEW RETURNED	24-Sep-2021

GENERAL COMMENTS	The study addresses a very relevant topic. The intervention should be more detailed for a better understanding. They should describe the strategies that will be used to reduce differences between therapists and between patients.
--

REVIEWER	Mehl, Stephanie Philipps-Universität Marburg
REVIEW RETURNED	12-Oct-2021

GENERAL COMMENTS	The present study is the presentation of a trial protocol of a study (TYPPEX study) in the primary care Improving Access to Psychological Therapies programme (IAPT). The authors did a really impressive job and described their planned trial in a clear-cut and impressive way. The protocol is elaborated and readers are able (based on the description and additional data) to understand all details and procedures of the trial. I have some small suggestions. First, I have to add that I am not familiar with the statistical procedures that are planned, thus, I recommend an additional review by a clinical trial statistician. The planned analyses sound highly adequate to me, but I have not performed them myself, thus, I am unsure. I have some small suggestions that I hope might improve the quality of the manuscript: Abstract: stepped-wedge cluster should be explained in more detail. A reader
---

	might ask himself/herself, how many therapists and patients will be enrolled in the trial? I believe this is one of the research questions of the trial and should be explained. Introduction: The literature review is a little bit confusing for me. It would be more helpful to include some literature on the efficacy and effectiveness of CBT for psychosis. And I believe the authors focus their trial not so much on patients with psychosis and the relevant diagnoses (schizophrenia, schizoaffective disorder), but on patients with anxiety and depression and subclinical psychotic experiences. It would be helpful to learn more on whether CBT is helpful in this subgroup with subclinical psychotic experiences. Also, it is unclear to me whether the authors already performed a feasibility trial on their therapy, they write about it in the methods section (page 8, line 25), but it is not cited. This trial should be explained in more detail in the introduction. Methods: It was unclear to me how many persons with lived experiences were involved in the trial ? Page 8, line 8: readers are not aware on how IAPT services work, this should be explained in slightly more detail. page 8, line 25: it is unclear whether the feasibility study exists ? page 9, line 22 readers might not know the definition of a trust page 9, line 58: I would like to learn more on the qualification of IAPT CBT therapists (study subject, CBT education) because this is different in many countries. page 10, line 4: why were therapists excluded who work in more than one local IAPT team? page 10, line 21: which mental disorders are associated with a referral to NICE step 4 or 5? page 11, line 16-17 I did not understand, how the difference between control and intervention condition of $d = 0.39-0.58$ was computed and would like to know more about this difference and the trial results it was based upon. I hope very much that my comments are helpful.
--	---

REVIEWER	Gold, Lisa Deakin University, Deakin Health Economics
REVIEW RETURNED	08-Mar-2022

GENERAL COMMENTS	This is a clear study protocol describing an in-progress study that has not completed recruitment. I only have minor suggestions for improvement of presentation of information for the reader - there is nothing missing from the protocol.  1. (p5 lines 19-21) Abstract: please add sample size to the abstract - just state the intended N of both therapists and service users 2. (p6 line 13) Strengths and limitations - dot point 2 - state how the study will “add to clinical understanding” 3. (p7 line 37) Methods - line 4 - participant timelines are not in Table 2, just delete “timelines and” here 4. (p8 line 55) number in UK (“1.3”) not Euro (“1,3”) to be consistent
--

	with rest of the paper 5. (p10 lines 13-33) The least-clear part of the protocol is the different ethics for the clinical and economic data. This makes sense to me as a researcher but it could be explained better. First, discuss all the clinical data first and start a new paragraph with "Additional.." (line 14). Clearly state here (p10 lines 10-14) that the consent process for all clinic-collected data (CAPE-P15, GAD-7, PHQ9 and ADSM) is an opt-out process. It would add value here to state what opt-out % you got in the feasibility study if you have that information available. Second, move the text at p10 lines 20-23 to the end of the section on clinical data (before the "Additional.." paragraph). Finally, in the "Additional..." paragraph, clarify that the "health economic sub-study" means the costs and quality of life data (perhaps refer to Table 2?) Actually I do have one question here - which is - why is "able and willing to provide written informed consent" a requirement for the opt-in ethics but not for the opt-out ethics? Shouldn't it be a criteria for ALL research participation? (I'll just leave that with you - the study has ethics approval already, but it just reads as if you are putting users through opt-out procedures who may not have the English ability or skills to understand their rights to opt-out. Maybe just be careful how you describe this in future publications). 6. (p10 line 37) Randomisation. I had a question here about exactly how therapists would be "grouped as study clusters" which is currently answered on p11 lines 25-28. Perhaps move the latter 2 sentences earlier to clarify how groups are formed when you first mention them. 7. (p11 line 3) Clarify that the caseload is 15-20 service users per (6-month) period. Ideally, also move up the line currently at p11 line 31-33 here too - to clarify that you will only include users who complete therapy during a period (I immediately had a question about how you will treat caseload that ran across periods, which this sentence answers). 8. (p11 line 16) If you mean that you are looking for a change in recovery rate from 0.39 (control) to 0.58 (intervention) can you perhaps state it like that 9. (p11 line 23) There is no reference [56] 10. (p12 line 10-16) Please state whether there is any reimbursement for the time of participants to complete the original data collection of the health economic sub-study. Presumably this adds up to an hour or two of time per person over the 5 rounds of data collection - ethically this should be reimbursed (at least with a gift voucher), and statistically if it is not reimbursed then you will have differential participation by ability to "donate" time to research (ie socio-economic bias) (especially with complete data analysis approaches) which will impact results. 11. (p12 line 34) Clarify "delivered" is "delivered face-to-face online or in-person" (if I have interpreted the covid-pivot in Table 1 correctly) 12. (p12 lines 36-) Please state if you are collecting data on the use of the different training aspects (workshop participation, use of
--	---

	online network, etc) 13. (p13 lines 15-17) I had a question here that the first intervention period would be like a “run-in” or “training” period - but this is answered by a sentence currently on p15 lines 23-26 - perhaps move this sentence to explain this vision of how the intervention is interpreted. 14. (p13 lines 54-59) The process evaluation aims are very broad (“assess views... investigate the implementation...”) for a very large proposed sample - could you add some detail here perhaps with examples of constructs to be explored or preliminary questions? 15. (p16 line 12) Define what you mean by complete data here - as there are 5 data points per person, but you mention interpolating QALY data, I was not clear if you will restrict analysis to 100% complete data (ie users who response to all 5 questionnaires) or will have some more lenient working definition of >50% or >70% complete? The stricter the definition of complete data analysis, the further the sub-sample gets from any sort of representation, with accompanying risks for validity of results (especially - as per point above - if you are relying on people donating their time to give data). 16. (p19-21) References - there is something odd in the formatting of references here, where there is no gap between the journal title and year? 17. (p26 lines 21-28) Table 3: (i) should the ** markers for the second footnote go on the PHQ-9 and GAD-7/ADSM?; and (ii) should there be an “X” in Screening for CAPE P-15 as the eligibility screen involves collection of CAPE P-15 data?
--	--

REVIEWER	Hunter, Rachael University College London, Research Dept of Primary Care and Population Health
REVIEW RETURNED	16-Mar-2022

GENERAL COMMENTS	The authors present a protocol for a step-wedge randomised control trial for Improving Access to Psychological Therapy (IAPT) therapists trained in Cognitive Behavioural Therapy tailoring for severity (CBT-ts) for individuals with common mental disorders and psychotic experiences. (CMD-PE). Overall the protocol is clear and well written, and presents an interesting study. I have the following queries: 1) I wasn't entirely clear how consent for collecting the primary outcome worked. I understand that the primary outcome is collected as part of routine care, but do you consent participants to collect the primary outcome? Or has ethical approval been obtained to collect the primary outcome without consent? 2) In relation to (1) is 12 month collection of the PHQ-9, GAD-7 and CAPE-P15 part of routine care? How complete is the routine collection of this data? The following questions relate to the economic evaluation only: 3) What descriptive statistics will be reported for participants that consent to providing the economic evaluation data to ensure it is a representative sample? 4) I'm a little concerned about the proposed validation of the EQ-5D 3L and 5L, particularly given the limited detail on the analysis. What analysis will be conducted to validate the two measures? Sensitive to changes in what? Recovery? I would like to see some more
---

	details on the psychometric analyses proposed included in here. 5) Again, for the 3L 5L validation, is there no concern about framing by providing the two side by side i.e. responses on the 5L may be tempered by responses on the 3L? I'd either like to see evidence that no framing occurs if the two are given together, or some consideration of additional measures being put in place, such as a delay between 3L and 5L being administered or a random sample given both or just 1 to test if any framing occurs. 6) Given that recovery is the primary outcome, it is strange that the ReQoL https://www.reqol.org.uk/p/overview.html was not considered given that it focuses on recovery in mental health. Is there a reason no condition specific measure for mental health was considered? 7) Page 16 line 27-30: in regards to the ICER, I didn't quite understand that sentence. Is it if the differences are statistically significant the ICER will be reported? Is this trying to say that the ICER won't be reported in dominate/dominated circumstances, as this isn't 100% clear from the wording. 8) Have the authors given any consideration to using a mapping algorithm from PHQ-9 /GAD-7 to EQ-5D-5L to generate QALYs as this would then result in a more complete sample for QALYs that could be reported alongside routine data? I am concerned about how representative the economic evaluation sample is going to be and if it will reflect the findings from the primary outcome. 9) I wasn't clear of the relevance of reference 29 and its placement as it is mostly about the role of baseline data, not about how the economic evaluation and clinical analysis should be aligned (although I'm curious if such a reference exists). I would have thought that a paper on step wedge trial analysis methodology or something relevant to economic evaluations, clustering or step-wedge would have been more suitable. 10) I would have liked to see more details on how missing data will be handled.
--	--

VERSION 1 – AUTHOR RESPONSE

Reviewer 1

Comment 1

The intervention should be more detailed for a better understanding.

Response: By addressing a number of comments from other reviewers below (See responses to comments 2 and 8 of Reviewer 2, and comments 13 and 14 of Reviewer3), we have added further detail about the content and delivery mechanisms for the therapy (CBT-ts). We have now included references of trials that used therapies that will inform CBT-ts (please see Introduction) and have clarified how CBT-ts training will be delivered, i.e. online.

Comment 2

They should describe the strategies that will be used to reduce differences between therapists and between patients.

Response: We aim to produce generalisable results for a broad clinical setting such as IAPT. Thus, we have not included criteria to reduce differences between therapists and patients beyond the ones included in our inclusion and exclusion criteria. Whilst this is an RCT, we have tried to mimic routine practice in IAPT services as much as possible with the purpose of a wide and rapid implementation of the intervention if the new, tailored psychological therapy proves to be effective for patients with common mental disorder and psychotic experiences, which currently represent almost 30% of their caseloads.

Reviewer 2

Comment 1

Abstract:

stepped-wedge cluster should be explained in more detail. A reader might ask himself/herself, how many therapists and patients will be enrolled in the trial? I believe this is one of the research questions of the trial and should be explained.

Response: The target number of therapists and service users has been added to the abstract. The stepped wedge design is referred to in the abstract and due to space constraints a detailed description is given in the main body of the paper (section Methods, Trial design and participants).

Comment 2

Introduction:

The literature review is a little bit confusing for me. It would be more helpful to include some literature on the efficacy and effectiveness of CBT for psychosis. And I believe the authors focus their trial not so much on patients with psychosis and the relevant diagnoses (schizophrenia, schizoaffective disorder), but on patients with anxiety and depression and subclinical psychotic experiences. It would be helpful to learn more on whether CBT is helpful in this subgroup with subclinical psychotic experiences.

Response: The text has been edited and references added to literature on the effectiveness of CBT for at-risk mental states:

1. Morrison AP, French P, Stewart SL, Birchwood M, Fowler D, Gumley AI, Jones PB, Bentall RP, Lewis SW, Murray GK, Patterson P, Brunet K, Conroy J, Parker S, Reilly T, Byrne R, Davies LM, Dunn G. Early detection and intervention evaluation for people at risk of psychosis: multisite randomised controlled trial. *BMJ*. 2012 (5)344:e2233.
2. van der Gaag M, Nieman DH, Rietdijk J, Dragt S, Ising HK, Klaassen RM, Koeter M, Cuijpers P, Wunderink L, Linszen DH. Cognitive behavioral therapy for subjects at ultrahigh risk for developing psychosis: a randomized controlled clinical trial. *Schizophr Bull*. 2012 Nov;38(6):1180-8.

Comment 3

Also, it is unclear to me whether the authors already performed a feasibility trial on their therapy, they write about it in the methods section (page 8, line 25), but it is not cited. This trial should be explained in more detail in the introduction.

Response: A brief summary of the feasibility study has been added to the introduction section. The study is unpublished but a study report is available on request.

Comment 4

Methods:

It was unclear to me how many persons with lived experiences were involved in the trial?

Response: The size of the Lived Experience Advisory Panel (11) is given in the text. Numbers have been added for the IAPT service user contributors (three) and IAPT therapist contributors (four) involved during the trial.

Comment 5

Page 8, line 8: readers are not aware on how IAPT services work, this should be explained in slightly more detail.

Response: A description of IAPT services is given in paragraph 2 of the introduction and in detail in the Study Setting section.

Comment 6

page 8, line 25: it is unclear whether the feasibility study exists?

Response: Reference to the feasibility study now added to the Introduction section, addressing the comment above.

Comment 7

page 9, line 22 readers might not know the definition of a trust

Response: A description of an NHS trust has been added

Comment 8

page 9, line 58: I would like to learn more on the qualification of IAPT CBT therapists (study subject, CBT education) because this is different in many countries.

Response: Sentences and a reference added to expand on therapist training: "Step 3 IAPT CBT therapists will have completed a one-year postgraduate diploma (PG Dip) in CBT accredited by the British Association for Behavioural and Cognitive Psychotherapies (BABCP). Entry requirements for the PG Dip include a secured work placement for clinical practice and a British Association for Behavioural and Cognitive Psychotherapies (BABCP) recognised core profession, such as mental health nursing or counselling psychology."

Comment 9

page 10, line 4: why were therapists excluded who work in more than one local IAPT team?

Response: Added the statement "(to avoid potential contamination e.g. via team meetings, shared office space)"

Comment 10

page 10, line 21: which mental disorders are associated with a referral to NICE step 4 or 5?

Response: Added phrase "complex and severe depression". We also removed step 5 as this was an error. There are only four steps in the NICE stepped care model.

Comment 11

page 11, line 16-17 I did not understand, how the difference between control and intervention condition of $d = 0.39-0.58$ was computed and would like to know more about this difference and the trial results it was based upon.

Response: This recovery rates estimates emerged from our previous work (Perez J, Russo DA, Stochl J, Clarke J, Martin Z, Jassi C, French P, Fowler D, Jones PB. Common mental disorder including psychotic experiences: Trailblazing a new recovery pathway within the Improving Access to Psychological Therapies programme in England. *Early Interv Psychiatry*. 2018 Jun;12(3):497-504. doi: 10.1111/eip.12434), which is referenced in that Section and was submitted to the NIHR as part of our successful grant application. In a subsequent replication study, with a much bigger sample, we found out that this estimation was conservative, given that the recovery rates would be closer to 0.27 for people with psychotic experiences and 0.62 for those without them (Knight C, Russo D, Stochl J, Croudace T, Fowler D, Grey N, Reeve N, Jones PB, Perez J. Prevalence of and recovery from common mental disorder including psychotic experiences in the UK Primary Care Improving Access to Psychological Therapies (IAPT) Programme. *J Affect Disord*. 2020 Jul 1;272:84-90. doi: 10.1016/j.jad.2020.04.015). However, we decided to stick to the first, most conservative estimation to make entirely sure that the trial is strongly powered.

Reviewer 3

Comment 1

(p5 lines 19-21) Abstract: please add sample size to the abstract - just state the intended N of both therapists and service users

Response: Added as above. See response to comment 1 of Reviewer 2.

Comment 2

(p6 line 13) Strengths and limitations - dot point 2 - state how the study will “add to clinical understanding”

Response: We have removed that sentence as we agree it is confusing.

Comment 3

(p7 line 37) Methods - line 4 - participant timelines are not in Table 2, just delete “timelines and” here

Response: The tables were numbered incorrectly and have been re-numbered. Table 2 contains participant timelines and measures, and Table 3 contains the study outcomes. The main text is therefore now correct.

Comment 4

(p8 line 55) number in UK (“1.3”) not Euro (“1,3”) to be consistent with rest of the paper

Response: Changed to 1.3.

Comment 5

(p10 lines 13-33) The least-clear part of the protocol is the different ethics for the clinical and economic data. This makes sense to me as a researcher but it could be explained better.

First, discuss all the clinical data first and start a new paragraph with “Additional..” (line 14). Clearly state here (p10 lines 10-14) that the consent process for all clinic-collected data (CAPE-P15, GAD-7,

PHQ9 and ADSM) is an opt-out process. It would add value here to state what opt-out % you got in the feasibility study if you have that information available. Second, move the text at p10 lines 20-23 to the end of the section on clinical data (before the “Additional..” paragraph).

Response: Text re-worked and clarified in line with your suggestions. With regards to opt-out % during feasibility, there were no trial-specific opt-outs.

Comment 6

Finally, in the “Additional...” paragraph, clarify that the “health economic sub-study” means the costs and quality of life data (perhaps refer to Table 2?)

Response Added clarification and reference to the table – now numbered Table 3.

Comment 7

Actually I do have one question here - which is - why is “able and willing to provide written informed consent” a requirement for the opt-in ethics but not for the opt-out ethics? Shouldn’t it be a criteria for ALL research participation? (I’ll just leave that with you - the study has ethics approval already, but it just reads as if you are putting users through opt-out procedures who may not have the English ability or skills to understand their rights to opt-out. Maybe just be careful how you describe this in future publications).

Response: In terms of language barriers, the opt-out participants are under the clinical care of an IAPT therapist, and NICE guidance relating to providing psychological treatment for people with depression and anxiety explicitly stipulates that services must provide, and therapists must work proficiently with, an independent interpreter if one is needed (NICE, 2009, 2011). Therefore patients have the ability to discuss and opt-out from all or trial-specific research data use. Patients unable to consent for other reasons, for example those with significant cognitive impairment, would be under the care of specialist services rather than IAPT. All IAPT patients have the ability to give informed consent and make decisions for themselves. Overall therefore, we feel that this is covered in terms of the protocol by the criteria “accepted onto the IAPT caseload for therapy and therefore meet service-specific inclusion criteria to access IAPT treatment”. However, you make a very important point and we will clarify and expand on this when reporting on the outcome of the trial.

Comment 8

(p10 line 37) Randomisation. I had a question here about exactly how therapists would be “grouped as study clusters” which is currently answered on p11 lines 25-28. Perhaps move the latter 2 sentences earlier to clarify how groups are formed when you first mention them.

Response: Sentences moved as suggested.

Comment 9

(p11 line 3) Clarify that the caseload is 15-20 service users per (6-month) period. Ideally, also move up the line currently at p11 line 31-33 here too - to clarify that you will only include users who complete therapy during a period (I immediately had a question about how you will treat caseload that ran across periods, which this sentence answers).

Response: Added “per six month period”.

We have re-worded the subsequent text to clarify that only control patients who finish therapy prior to their cluster receiving training will be included, but also that only intervention patients who finish therapy prior to the end of the intervention phase (i.e. end of step 4) will be included.

Comment 10

(p11 line 16) If you mean that you are looking for a change in recovery rate from 0.39 (control) to 0.58 (intervention) can you perhaps state it like that

Response: Edited as suggested.

Comment 11

(p11 line 23) There is no reference [56]

This reference was omitted in error and has now been renumbered and included: Hussey MA, Hughes JP. Design and analysis of stepped wedge cluster randomized trials. *Contemp Clin Trials*. 2007 Feb;28(2):182-91.

Comment 12

(p12 line 10-16) Please state whether there is any reimbursement for the time of participants to complete the original data collection of the health economic sub-study. Presumably this adds up to an hour or two of time per person over the 5 rounds of data collection - ethically this should be reimbursed (at least with a gift voucher), and statistically if it is not reimbursed then you will have differential participation by ability to "donate" time to research (ie socio-economic bias) (especially with complete data analysis approaches) which will impact results.

Response: Due to the sample size of 300 participants, and 5 follow-up timepoints, reimbursement would be a sizeable cost to the study and is not typical for NIHR-funded research unless there is a particular concern about representativeness in the target population or issues with data completion that merit an incentivisation approach. The questionnaires in the TYPPEX battery are very straightforward (e.g. EQ-5D) and in this case the independent steering committee and NHS ethics committee were happy with the protocol as described. We will however discuss this as a potential limitation of the health economic evaluation when reporting on the trial results.

Comment 13

(p12 line 34) Clarify "delivered" is "delivered face-to-face online or in-person" (if I have interpreted the covid-pivot in Table 1 correctly)

Response: Edited to clarify that the training is delivered online, and edited further down to clarify that supervision is delivered either online or face-to-face. Please note that the amendment referred to in Table 1 indicates the training can be delivered either online or face-to-face. However for practical reasons we are now likely to continue with the online model.

Comment 14

(p12 lines 36-) Please state if you are collecting data on the use of the different training aspects (workshop participation, use of online network, etc)

Response: We are now collecting data relating to training attendance and use of Resources, as part of our assessment of therapists' overall adherence and engagement with CBT-ts principles.

This was addressed in a ethics amendment dated 28/06/2021 now added to Table 1, and the following text has been added to the section indicated by the reviewer:

"Trial data collection on therapist participants will include training attendance, reasons for non-attendance, and implementation of the resources and components of the CBT-ts 'toolkit' with therapists' CAPE+ caseload during the supervision period. Therapists' will be assessed on their overall adherence and engagement with CBT-ts."

Comment 15

(p13 lines 15-17) I had a question here that the first intervention period would be like a “run-in” or “training” period - but this is answered by a sentence currently on p15 lines 23-26 - perhaps move this sentence to explain this vision of how the intervention is interpreted.

Response: We believe it is better to leave the sentence where it is as otherwise would alter the flow of the text. If the reviewer strongly feels that we should change it we would try our best to adjust it.

Comment 16

(p13 lines 54-59) The process evaluation aims are very broad (“assess views... investigate the implementation...”) for a very large proposed sample - could you add some detail here perhaps with examples of constructs to be explored or preliminary questions?

Response: Added Table 5 to indicate the key themes explored with each of the stakeholder groups interviewed in the process evaluation.

Comment 17

(p16 line 12) Define what you mean by complete data here - as there are 5 data points per person, but you mention interpolating QALY data, I was not clear if you will restrict analysis to 100% complete data (ie users who response to all 5 questionnaires) or will have some more lenient working definition of >50% or >70% complete? The stricter the definition of complete data analysis, the further the sub-sample gets from any sort of representation, with accompanying risks for validity of results (especially - as per point above - if you are relying on people donating their time to give data).

Response: Yes, we are restricting analysis to people with 100% complete data. Interpolation is only possible for complete data. However, we are allowing EI-ADSUS and EQ-5D data to be collected for multiple time points if a time point is missed, thus reducing the amount of people excluded from the complete case analysis. Further, we will explore the impact of missing data on the results as described in the paper: “Sensitivity analyses will be conducted to explore the impact of missing data (using multiple imputation)... Additionally, to assess the impact of missing data, we will compare those with economic data to the full sample included in the main clinical analysis to examine any potential biases in terms of demographic and clinical factors.”

Comment 18

(p19-21) References - there is something odd in the formatting of references here, where there is no gap between the journal title and year?

Response: Edited accordingly.

Comment 19

*17. (p26 lines 21-28) Table 3: (i) should the ** markers for the second footnote go on the PHQ-9 and GAD-7/ADSM?; and (ii) should there be an “X” in Screening for CAPE P-15 as the eligibility screen involves collection of CAPE P-15 data?*

Response: That is correct and we have edited accordingly.

Reviewer 4

Comment 1

- 1) *I wasn't entirely clear how consent for collecting the primary outcome worked. I understand that the primary outcome is collected as part of routine care, but do you consent participants to collect the primary outcome? Or has ethical approval been obtained to collect the primary outcome without consent?*

Response: we agree that the original wording was unclear; the section "Ethics and dissemination" has been re-written to clarify the nature of the consent. Patients are not consented to collect the primary outcome as it is collected during routine care and de-identified prior to sharing with the research team. Patients are able to opt-out of this data collection, and ethical approval has been granted for this approach.

- 2) *In relation to (1) is 12 month collection of the PHQ-9, GAD-7 and CAPE-P15 part of routine care? How complete is the routine collection of this data?*

Response: The 12 month collection of clinical measures is not part of routine collection and occurs outside of treatment; patients are unlikely to still be receiving care from IAPT at 12 months after baseline. Therefore 12 month follow-up is only collected for those patients who consented to provide data for the economic evaluation. This point is clarified in the edits to the text in "Ethics and dissemination".

Comment 2

The following questions relate to the economic evaluation only:

Response: Apologies for the lack of detail on the economic methods. There is limited space in the protocol so we could not include as much detail as we would have liked. We have added detail here for your information but also made additions to the paper.

Comment 3

What descriptive statistics will be reported for participants that consent to providing the economic evaluation data to ensure it is a representative sample?

Response: Before we conduct any economic analyses we will compare those with economic data to the full sample included in the main clinical analysis to examine any potential biases. This will include a range of important demographic and clinical factors. We have added this to the paper: *"Additionally, to assess the impact of missing data, we will compare those with economic data to the full sample included in the main clinical analysis to examine any potential biases in terms of demographic and clinical factors."*

Comment 4

I'm a little concerned about the proposed validation of the EQ-5D 3L and 5L, particularly given the limited detail on the analysis. What analysis will be conducted to validate the two measures? Sensitive to changes in what? Recovery? I would like to see some more details on the psychometric analyses proposed included in here.

Response: We have not included any details on the EQ-5D-3L and 5L comparison as this is not technically part of the trial. This comparison will be conducted before the economic evaluation of the trial is conducted to inform us on which outcome to use.

The final psychometric analysis is not yet finalised as we are examining what the most up to date approaches are but it is likely we will include concurrent validity, convergent validity, known-group validity and responsiveness.

Comment 5

Again, for the 3L 5L validation, is there no concern about framing by providing the two side by side i.e. responses on the 5L may be tempered by responses on the 3L? I'd either like to see evidence that no framing occurs if the two are given together, or some consideration of additional measures being put in place, such as a delay between 3L and 5L being administered or a random sample given both or just 1 to test if any framing occurs.

Response: We had not considered the issue of framing and now are unlikely to be able to account for this. We will include this as a limitation of the EQ-5D comparison.

Comment 6

Given that recovery is the primary outcome, it is strange that the ReQoL <https://eur01.safelinks.protection.outlook.com/?url=https%3A%2F%2Fwww.reqol.org.uk%2Fp%2Foverview.html&data=04%7C01%7CP.Ashford%40uea.ac.uk%7C74103734ddad471bd28608da0a5ec038%7Cc65f8795ba3d43518a070865e5d8f090%7C0%7C0%7C637833699313830273%7CUnknown%7CTWFpbGZsb3d8eyJWIjoiMC4wLjAwMDAiLCJQIjoiV2luMzliLCJBTiI6IjEhaWwiLCJXVCi6Mn0%3D%7C3000&data=PzqxYr9uNdaaS1QcMkRIZ0uzncnpjXHJ9Beav4jRRIU%3D&mp:reserved=0> was not considered given that it focuses on recovery in mental health. Is there a reason no condition specific measure for mental health was considered?

Response: We agree it would have been interesting to include the ReQoL in this study. However, at the time of writing the grant application (which was awarded in April 2017), the ReQoL did not exist. Further, the ReQoL is not currently recommended by NICE even for projects on mental health, and they prefer the EQ5D: National Institute for Health and Care Excellence. NICE health technology evaluations: the draft manual. 2021. <https://www.nice.org.uk/Media/Default/About/what-we-doour-programmes/nice-guidance/chte-methods-and-processes-consultation/health-technology-evaluations-manual.docx> Therefore, we chose to use the EQ-5D.

Comment 7

Page 16 line 27-30: in regards to the ICER, I didn't quite understand that sentence. Is it if the differences are statistically significant the ICER will be reported? Is this trying to say that the ICER won't be reported in dominate/dominated circumstances, as this isn't 100% clear from the wording.

Response: This sentence is misleading. We have replaced "Incremental cost-effectiveness ratios (ICERs) will be calculated if higher costs and better outcomes are found in either the intervention or control group" with "Incremental cost-effectiveness ratios (ICERs) will be calculated".

Comment 8

Have the authors given any consideration to using a mapping algorithm from PHQ-9 /GAD-7 to EQ-5D-5L to generate QALYs as this would then result in a more complete sample for QALYs that could be reported alongside routine data? I am concerned about how representative the economic evaluation sample is going to be and if it will reflect the findings from the primary outcome.

Response: We are including the PHQ-9 and GAD-7 as part of the IAPT-defined recovery composite measure from the primary clinical analysis in a cost-effectiveness analysis. Even if we did map PHQ-9 and GAD7 to the EQ-5D to generate QALYs, this would not increase numbers in the economic analysis as we would still be missing resource use data and therefore costs for anyone who did not complete the economic measures. Therefore this approach would provide no additional benefits.

Comment 9

I wasn't clear of the relevance of reference 29 and its placement as it is mostly about the role of baseline data, not about how the economic evaluation and clinical analysis should be aligned

(although I'm curious if such a reference exists). I would have thought that a paper on step wedge trial analysis methodology or something relevant to economic evaluations, clustering or step-wedge would have been more suitable.

Response: This reference is related to the decision to control for baseline differences between the groups. We have moved the reference so that is clearer.

Comment 10

I would have liked to see more details on how missing data will be handled.

Response: We have added detail as per the above on comparing those with and without economic data to explore potential biases. Additionally, we will rerun the base case analysis using multiple imputation to account for missing data and compare the results to the base case. We have added this in the paper: "Sensitivity analyses will be conducted to explore the impact of missing data (using multiple imputation)".

VERSION 2 – REVIEW

REVIEWER	Gold, Lisa Deakin University, Deakin Health Economics
REVIEW RETURNED	11-May-2022

GENERAL COMMENTS	All comments raised in original review have been suitably discussed and/or addressed - thank you.
---

REVIEWER	Hunter, Rachael University College London, Research Dept of Primary Care and Population Health
REVIEW RETURNED	12-May-2022

GENERAL COMMENTS	1) Apologies, but I still don't understand the data collection schedule in Table 2. Based on the author response service users won't still be in treatment at 12 months, but there is a 12-month data collection point for PHQ-9, GAD-7 and CAPE P-15. How is this data being collected? 2) The response to comment 8 (mapping algorithms) highlights that some additional information is needed for the health economic analysis. Based on the methods for the economic analysis described in the paper (adjustment for stepped wedge design) a two-stage bootstrap is the most likely method required to combine costs and QALYs. If the two-stage bootstrap is used then an imbalance in numbers on costs and outcomes is possible - you can include outcome data for people with missing cost data. This is commonly done in the reverse circumstance where routine resource use data is available, but no outcome data. The actual reason why this would not be fruitful is due to the answer to (1) above (there is likely to be equivalent missing data at 12 months for the PHQ-9 and GAD-7). 3) I still think more can be done with the routine data to check that the results of the health economic analysis aren't biased due to the data collection method, for example a 12 week economic evaluation using the routine data and checking the results with that from the sample. 4) Just to note, in response to comment 6, the NICE guidance doesn't recommend against the ReQoL. Instead it suggests a hierarchy of evidence (see Figure 4.1) with the EQ-5D being preferred, but if there is evidence it is not appropriate another
---

	outcome measure can be used, with a preference based condition specific measure falling after generic measures in terms of hierarchy. Although there is good evidence for the EQ-5D in common mental health problems, it is less clear for the population in this study as the EQ-5D has been argued against in more severe psychosis. That the authors are suggesting to test the validity of the EQ-5D would suggest that they agree there is an evidence gap for this group and it's not clear to what extent it is a suitable measure. I appreciate that the timing unfortunately was not right for the ReQoL, but the CORE-OM/CORE-6D as a preference based mental health measure could have been considered instead.
--	---

VERSION 2 – AUTHOR RESPONSE

Comment 1

Apologies, but I still don't understand the data collection schedule in Table 2. Based on the author response service users won't still be in treatment at 12 months, but there is a 12-month data collection point for PHQ-9, GAD-7 and CAPE P-15. How is this data being collected?

Response

The process to collect measures at 12 months is explained in the text under the subheading “Ethics and Dissemination”. However, we have also included it in Table 2’s footnote: “At 12 month follow-up, these measures will be collected via an opt-in process to provide additional data.”

Comment 2

The response to comment 8 (mapping algorithms) highlights that some additional information is needed for the health economic analysis. Based on the methods for the economic analysis described in the paper (adjustment for stepped wedge design) a two-stage bootstrap is the most likely method required to combine costs and QALYs. If the two-stage bootstrap is used then an imbalance in numbers on costs and outcomes is possible - you can include outcome data for people with missing cost data. This is commonly done in the reverse circumstance where routine resource use data is available, but no outcome data. The actual reason why this would not be fruitful is due to the answer to (1) above (there is likely to be equivalent missing data at 12 months for the PHQ-9 and GAD-7).

Response

As the reviewer states, there is likely to be missing data as 12 months for the PhQ-9 and GAD-7 therefore this approach is unlikely to be useful.

Comment 3

I still think more can be done with the routine data to check that the results of the health economic analysis aren't biased due to the data collection method, for example a 12 week economic evaluation using the routine data and checking the results with that from the sample.

Response

In the paper we have said: “Additionally, to assess the impact of missing data, we will compare those with economic data to the full sample included in the main clinical analysis to examine any potential biases in terms of demographic and clinical factors.” This will include using routinely collected data from baseline and follow-up and will highlight potential biases due to the data collection method.

Your suggestion to conduct a 12 week economic evaluation using routine data to check the results is an interesting one. We will consider conducting doing this in sensitivity analyses as part of our assessment of missing data.

Comment 4

Just to note, in response to comment 6, the NICE guidance doesn't recommend against the ReQoL. Instead it suggests a hierarchy of evidence (see Figure 4.1) with the EQ-5D being preferred, but if there is evidence it is not appropriate another outcome measure can be used, with a preference based condition specific measure falling after generic measures in terms of hierarchy. Although there is good evidence for the EQ-5D in common mental health problems, it is less clear for the population in this study as the EQ-5D has been argued against in more severe psychosis. That the authors are suggesting to test the validity of the EQ-5D would suggest that they agree there is an evidence gap for this group and it's not clear to what extent it is a suitable measure. I appreciate that the timing unfortunately was not right for the ReQoL, but the CORE-OM/CORE-6D as a preference based mental health measure could have been considered instead.

Response:

We recognise the limitation of the EQ-5D, thus we have included a CEA based on the clinical measures. While we could have included other measures, it is too late to do so now. We will raise the limitations of the EQ-5D in the any discussion of the results.